# Obesity-Related Indices Are Associated with Peripheral Artery Occlusive Disease in Patients with Type 2 Diabetes Mellitus

**DOI:** 10.3390/jpm11060533

**Published:** 2021-06-09

**Authors:** Chih-Hsuan Wung, Mei-Yueh Lee, Pei-Yu Wu, Jiun-Chi Huang, Szu-Chia Chen

**Affiliations:** 1Department of General Medicine, Kaohsiung Medical University Hospital, Kaohsiung 807, Taiwan; knash031130@hotmail.com.tw; 2Division of Endocrinology and Metabolism, Department of Internal Medicine, Kaohsiung Medical University Hospital, Kaohsiung Medical University, Kaohsiung 807, Taiwan; lovellelee@hotmail.com; 3Faculty of Medicine, College of Medicine, Kaohsiung Medical University, Kaohsiung 807, Taiwan; karajan77@gmail.com; 4Division of Nephrology, Department of Internal Medicine, Kaohsiung Medical University Hospital, Kaohsiung Medical University, Kaohsiung 807, Taiwan; wpuw17@gmail.com; 5Department of Internal Medicine, Kaohsiung Municipal Siaogang Hospital, Kaohsiung Medical University, Kaohsiung 812, Taiwan

**Keywords:** obesity-related index, diabetes mellitus, peripheral artery occlusive disease

## Abstract

Type 2 diabetes mellitus (DM) is an increasing global health issue. Peripheral artery occlusive disease (PAOD) is a common complication of diabetes, and it is a complex and costly disease. The association between type 2 DM and obesity is well known, however, the relationship between obesity and PAOD in patients with type 2 DM has yet to be elucidated. Therefore, the aim of this study was to examine associations between obesity-related indices and PAOD in patients with type 2 DM. A total of 1872 outpatients with type 2 DM were recruited from two hospitals in southern Taiwan. An ankle–brachial index (ABI) < 0.9 in either leg was considered to indicate the presence of PAOD. The following obesity-related indices were investigated: conicity index (CI), waist–hip ratio (WHR), body roundness index (BRI), waist-to-height ratio (WHtR), abdominal volume index, a body shape index (ABSI), visceral adiposity index (VAI), lipid accumulation product (LAP), body adiposity index, body mass index and triglyceride–glucose index. Overall, 4.1% of the enrolled patients had an ABI < 0.9. High values of the following obesity-related indices were significantly associated with a low ABI: WHtR (*p* = 0.045), VAI (*p* = 0.003), CI (*p* = 0.042), BRI (*p* = 0.021) and ABSI (*p* = 0.043). Furthermore, WHR (area under the curve (AUC) = 0.661), CI (AUC = 0.660) and LAP (AUC = 0.642) had the best performance (all *p* < 0.001) to predict PAOD. In conclusion, high WHtR, BRI, CI, VAI and BAI values were associated with a low ABI in the enrolled patients, and WHR, CI and LAP were the most powerful predictors of PAOD.

## 1. Introduction

Type 2 diabetes mellitus (DM) is a heterogeneous abnormality characterized by insulin resistance and defective insulin secretion [1]. A reported 451 million people had diabetes globally in 2017, a number which is expected to increase to 629 million by 2045. In addition, the prevalence of DM in people aged 18–99 years in 2017 was 8.4%, and this has been predicted to increase to 9.9% by 2045 [2]. In Taiwan, the age-standardized prevalence of DM in people aged 20–79 years was 6.45% in 2014 [3]. The increasing prevalence of DM and DM-related complications worldwide may increase the pressure on healthcare systems [4]. DM-associated complications include both macrovascular and microvascular diseases. The macrovascular diseases include peripheral artery occlusive disease (PAOD), cerebrovascular disease and coronary heart disease, and the microvascular diseases include neuropathy, nephropathy and retinopathy [5]. PAOD affects 9–24% of patients with DM, of whom 8–25% will develop diabetic foot ulcers [6]. The ankle–brachial index (ABI) is a commonly used noninvasive tool to quantitatively evaluate arterial health with regard to arterial stiffness and blockage [7]. The ABI has been shown to be a reliable marker of atherosclerosis, with a lower value being associated with generalized atherosclerosis. Specifically, an ABI < 0.9 is commonly used to diagnose PAOD clinically and in epidemiologic studies [8]. Previous studies have reported a significant association of ABI and the thickness of the intima media and common carotid artery with the degree of stenosis in the middle cerebral and intracranial internal carotid arteries [9]. Therefore, ABI and carotid intima–media thickness measurements may be good prognostic atherosclerosis markers for stroke [10]. Since PAOD increases the risk of nonhealing ulcers, infection and amputation, it is important to conduct periodic screening and foot care education for patients with DM [11].

Anthropometric indices, such as conicity index (CI), waist-to-height ratio (WHtR), waist–hip ratio (WHR), body roundness index (BRI), visceral adiposity index (VAI), abdominal volume index (AVI), lipid accumulation product (LAP), a body shape index (ABSI), body adiposity index (BAI), body mass index (BMI) and triglyceride–glucose index (TyG), can easily be calculated and quantified using values of triglycerides (TGs), body height (BH), hip circumference (HC), waist circumference (WC), high-density lipoprotein cholesterol (HDL-C) and body weight (BW) [12,13,14,15]. Previous epidemiologic studies have suggested a significant association between obesity and PAOD. In a report from Northern Nigeria, Agboghoroma et al. found that patients with both type 2 DM and peripheral artery disease were associated with high BMI and low HDL-C [16]. In addition, Jakovljevic et al. reported that high BMI, high WC, high WHR, fat tissue and visceral distribution were associated with aortoiliac peripheral arterial diseases in middle-aged men [17]. However, Agarwal et al. did not find a correlation between obesity and peripheral artery disease among patients with diabetes [18]. Therefore, the association between obesity and PAOD remains controversial. The aim of this study was to examine the association between obesity and PAOD through obesity-related indices among a diabetic population. In this study, we enrolled 1872 diabetic patients residing in the south of Taiwan and investigated the associations between PAOD defined according to ABI and the aforementioned obesity-related indices.

## 2. Materials and Methods

### 2.1. Study Patients

All outpatients aged more than 18 years with type 2 DM who visited two hospitals in southern Taiwan were enrolled in this study. The following patients were excluded: (1) those who were receiving dialysis; (2) those who had had undergone a renal transplantation; (3) those with type 1 DM (defined as the continued need for insulin treatment for at least 1 year after the diagnosis, the presence of diabetic ketoacidosis, ketonuria (≥3) or symptoms of severe hyperglycemia), (4) those who had neoplastic disease; and (5) those with critical ischemia conditions, such as pain, paralysis, paresthesia, pulselessness and paleness. Finally, 1872 patients (808 males and 1064 females; mean age 64.0 ± 11.3 years) were included in this study. All of the included patients provided written informed consent to participate in this study, and the study protocol was approved by the Institutional Review Board of Kaohsiung Medical University Hospital. In addition, this study was conducted in accordance with the relevant guidelines.

### 2.2. Medical, Laboratory and Demographic Data

All of the patients were interviewed and their medical records reviewed, after which their medical and demographic data, including sex, age and co-morbidities, were recorded. Fasting blood samples were obtained from all of the patients, and laboratory tests were conducted using a COBAS Integra 400 autoanalyzer (Roche Diagnostics GmbH, D-68298, Mannheim, Germany), which was also used to measure serum creatinine levels with the compensated Jaffé (kinetic alkaline picrate) method using a calibrator traceable to isotope dilution mass spectrometry [19]. Laboratory data, including fasting glucose, HbA_1c_, TGs, total cholesterol, HDL-C and low-density lipoprotein cholesterol (LDL-C), were obtained from the fasting blood samples. Estimated glomerular filtration rate (eGFR) values were calculated using the Chronic Kidney Disease Epidemiology Collaboration equation [20]. In addition, data regarding prescriptions of the following drugs were obtained from medical records: angiotensin-converting enzyme inhibitors (ACEIs), statins and angiotensin II receptor blockers (ARBs).

### 2.3. ABI and Definition of PAOD

ABI was measured once in each patient using a non-invasive vascular screening device (VP1000; Collin Co. Ltd., Komaki, Japan) at the diabetes outpatient clinics by a trained diabetic nurse [21,22,23]. Before ABI was measured, study patients were instructed to lie quietly and breathe normally in the supine position for at least 10 min. The device measured the blood pressures in the ankles and arms, and ABI was calculated as: systolic blood pressure (SBP) in the ankles/SBP in the arms. PAOD was defined as an ABI < 0.9 in either leg. The reproducibility and validity of this ABI-form device have been described previously [22].

### 2.4. Calculation of Obesity-Related Indices

BMI = BW (kg)/BH2 (m).

WHR = WC (cm)/HC (cm).

WHtR = WC (cm)/BH (cm).

LAP = $\left({\mathrm{WC}}_{\left(\mathrm{cm}\right)}-65\right)\times {\;\mathrm{TG}}_{\left(\mathrm{mmol}/L\right)}$ in males, and

LAP = $\left({\mathrm{WC}}_{\left(\mathrm{cm}\right)}-58\right)\times {\;\mathrm{TG}}_{\left(\mathrm{mmol}/L\right)}$ in females [24].

BRI = $364.2-365.5\times \sqrt{1-{\left(\frac{\frac{{\mathrm{WC}}_{\left(m\right)}}{2\pi }}{0.5\times {\mathrm{BH}}_{\left(m\right)}}\right)}^{2}}$ [25].

CI = $\frac{{\mathrm{WC}}_{\left(m\right)}}{0.109\times \sqrt{\frac{{\mathrm{BW}}_{\left(\mathrm{kg}\right)}}{{\mathrm{BH}}_{\left(m\right)}}}}$ [26].

VAI = $\left(\frac{{\mathrm{WC}}_{\left(\mathrm{cm}\right)}}{39.68+\left(1.88\times \;\mathrm{BMI}\right)}\right)\times \left(\frac{{\mathrm{TG}}_{\left(\mathrm{mmol}/L\right)}}{1.03}\right)\times \left(\frac{1.31}{{\mathrm{HDL}}_{\left(\mathrm{mmol}/L\right)}}\right)$ in males, and

VAI = $\left(\frac{{\mathrm{WC}}_{\left(\mathrm{cm}\right)}}{36.58+\left(1.89\times \;\mathrm{BMI}\right)}\right)\times \left(\frac{{\mathrm{TG}}_{\left(\mathrm{mmol}/L\right)}}{0.81}\right)\times \left(\frac{1.52}{{\mathrm{HDL}}_{\left(\mathrm{mmol}/L\right)}}\right)$ in females [27].

BAI = $\frac{{\mathrm{HC}}_{\left(\mathrm{cm}\right)}}{{\mathrm{BH}}_{\left(m\right)}{}^{3/2}}-18$ [28].

AVI = $\frac{2\times {\left({\;\mathrm{WC}}_{\left(\mathrm{cm}\right)}\;\right)}^{2}+0.7\times {\left({\;\mathrm{WC}}_{\left(\mathrm{cm}\right)}-{\mathrm{HC}\;}_{\left(\mathrm{cm}\right)}\right)}^{2}}{1000}$ [29].

ABSI = WC (m)/[BMI2/3(kg/m^2^) × BH1/2(m)] [30].

TyG index = Ln [fasting TG (mg/dL) × fasting plasma glucose (mg/dL)/2] [31].

### 2.5. Statistical Analysis

Data are expressed as a percentage, mean ± standard deviation or median (25th–75th percentile) for TGs. Between-group differences in categorical variables were analyzed using the chi-square test, and differences in continuous variables were analyzed using the independent *t*-test. Linear regression analysis was used to identify the association between obesity-related indices and ABI. Age, sex and significant variables in univariate analysis were selected for multivariate analysis. The predictive performances of the studied obesity-related indices for an ABI < 0.9 were analyzed using receiver operating characteristic (ROC) curves. Furthermore, the predictive abilities of the indices were assessed using areas under the ROC curves (AUCs). *p* values < 0.05 were considered to be statistically significant. Statistical analyses were conducted using SPSS for Windows version 26.0 (SPSS Inc., Chicago, IL, USA).

## 3. Results

Overall, 4.1% of the enrolled patients had an ABI < 0.9. The mean age of the 1872 enrolled patients was 64.0 ± 11.3 years, and included 1064 females and 808 males.

### 3.1. Comparisons of Baseline Characteristics between the Patients with an ABI ≥ 0.9 and <0.9

Table 1 shows comparisons of the clinical characteristics of the patients with an ABI ≥ 0.9 and < 0.9. The ABI < 0.9 group was older, were prescribed with more ACEIs, ARBs and statins and had higher TGs and WC and lower ABI, eGFR and HDL-C than the ABI ≥ 0.9 group. In addition, the ABI < 0.9 group had a higher VAI, WHR, WHtR, LAP, AVI, ABSI and BRI than the ABI ≥ 0.9 group.

### 3.2. Determinants of ABI

The results of univariable linear regression analysis showed that old age, high TGs, high total cholesterol, low HDL cholesterol, high LDL cholesterol, low eGFR and the use of ACEIs/ARBs and statins were associated with low ABI (Table 2).

Table 3 shows the results of multivariable linear regression analysis for the determinants of ABI in the study patients. The following multivariable linear regression analyses were performed for the different indices:

Model 1. Adjusted for age, sex, log TGs, total cholesterol, HDL-C, LDL-C, eGFR and the use of medications for AVI, BMI, BAI, BRI, WHtR, WHR, CI, ABSI and TyG index.

Model 2. Adjusted for age, sex, total cholesterol, HDL-C, LDL-C and the use of medications for LAP.

Model 3. Adjusted for age, sex, total cholesterol, LDL-C and the use of medications for VAI.

The results of these analyses showed that high values of the following indices were significantly associated with a low ABI: WHtR (per 0.1; coefficient β, –0.009; *p* = 0.045), BRI (per 1; β, –0.005; *p* = 0.021), CI (per 0.1; β, –0.007; *p* = 0.042), VAI (per 1; β, –0.003; *p* = 0.003) and ABSI (per 1; β, –0.010; *p* = 0.043). However, the other indices (BMI, WHR, LAP, BAI, AVI, TyG index) were not significantly associated with ABI.

### 3.3. The Predictive Ability of the Obesity-Related Indices to Identify an ABI < 0.9

The ROC and AUC analyses, cutoff values, sensitivity, specificity and Youden index of the 11 studied indices in identifying an ABI < 0.9 are shown in Table 4. The AUCs for the indices in descending order were WHR (0.661, *p* < 0.001), CI (0.660, *p* < 0.001), LAP (0.642, *p* < 0.001), ABSI (0.638, *p* < 0.001), VAI (0.633, *p* < 0.001), WHtR and BRI (0.630, *p* < 0.001), AVI (0.614, *p* = 0.001), TyG index (0.559, *p* = 0.078), BMI (0.547, *p* = 0.166) and BAI (0.527, *p* = 0.420).

## 4. Discussion

In this study of 1872 patients with type 2 DM, we investigated the relationships between ABI and 11 obesity-related indices, and found that high WHtR, BRI, CI, VAI and ABSI were significantly associated with a low ABI. Furthermore, WHR, CI and LAP were the three most powerful predictors of an ABI < 0.9.

The prevalence of obesity has increased in most Asian countries for almost two decades [32]. Several studies have shown a significant relationship between obesity, as assessed using BMI, and the risk of peripheral artery disease [33]. Vasheghani-Farahani et al. reported significant upward correlations between ABI and anthropometric indices including BW, WC, WHtR and BMI among premenopausal women aged 25 to 55 years [34]. In addition, Yeboah et al. reported that BMI and WC were higher in Ghanaian patients with peripheral artery disease [35]. Furthermore, in a hospital-based, cross-sectional study of 146 diabetic patients conducted in India, Agarwal et al. found that old age, long duration of diabetes, high SBP and DBP, smoking, higher glycated hemoglobin levels and coronary artery disease were significantly correlated with an ABI < 0.9, whereas obesity was not [18]. Our results showed that high values of WHtR, BRI, CI, VAI and ABSI were associated with a low ABI. The result of WHtR in our study was similar to that of Vasheghani-Farahani et al., indicating the importance of truncal fat [34]. BRI, CI, VAI and ABSI are simple indices which reflect abdominal fat and are used to assess central obesity [36]. One possible explanation for our findings may be related to leptin, an adipokine that regulates insulin secretion, lipid metabolism and hematopoiesis [37]. Lai et al. reported that patients with type 2 DM were associated with central obesity, higher leptin levels and worse cardiovascular autonomic neuropathy. In addition, a high leptin level has been associated with peripheral artery disease [38]. However, the detailed mechanisms by which leptin regulates these factors are unclear. Another possible explanation is adiponectin, an adipokine that plays important and protective roles in the regulation of glucose and lipid metabolism. Adiponectin has been shown to have anti-diabetic, anti-atherosclerotic and anti-inflammatory effects, however, the level of adiponectin has been reported to be lower in obese individuals and in those with diabetes and vascular disease [39]. Moreover, Gherman et al. reported that an elevated leptin level and decreased adiponectin level were associated with PAOD in a case–control cohort. Collectively, abnormal leptin and adiponectin levels are probably associated with BRI, CI, VAI and ABSI, indicators of central obesity and insulin resistance, leading to diabetes-induced PAOD; however, the detailed molecular mechanisms involved in these associations are unclear, and further studies are warranted.

Several studies have investigated the predictive ability of obesity-related indices for peripheral artery disease [16,17,34,35]. Planas et al. reported that abdominal fat and high WHR were associated with PAOD [40], and Rao et al. reported that a high WHR and obesity increased the risk of peripheral artery disease among patients with diabetes [41]. In addition, Wakabayashi et al. found that LAP was significantly correlated with a low ABI after treadmill exercise [42]. In our study, WHR, CI and LAP were the three most powerful predictors of PAOD among the patients with diabetes. Katsilambros et al. investigated the relationship between WHR and PAOD in diabetic patients [43], and reported findings consistent with our study. The authors concluded that this observation was in accordance with the concept that central obesity is related to atherogenicity [43]. CI is used to assess central obesity, and it has been associated with cardiovascular diseases [44]. Numerous studies have investigated the relationship between CI and cardiovascular diseases [45,46,47,48]. In the present study, we demonstrated the predictive ability of CI in identifying PAOD among diabetic patients. Abdominal fat has higher secretion and metabolic activity than peripheral fat [49], and both central obesity and abdominal fat accumulation can impair endothelial function, subsequently leading to the development of vascular diseases [17]. LAP is also a powerful index to assess abdominal fat. Visceral fat has a higher rate of lipolysis and free fatty acid production than subcutaneous fat, and an increase in free fatty acid levels could induce endothelial dysfunction and contribute to vascular diseases [50]. Further studies are needed to investigate the predictive abilities of WHR, CI and LAP in identifying PAOD among different populations.

In this study, high WHtR, VAI, BRI, CI and ABSI were associated with a low ABI, and WHR, WHtR, AVI, VAI, LAP, CI, BRI and ABSI were associated with an ABI < 0.9. However, other obesity-related indices, including BMI, BAI and TyG index, were not associated with ABI. WC is directly related to abdominal size. WHR, WHtR, LAP, BRI, CI, VAI, AVI and ABSI, calculated with WC, were used to assess abdominal obesity and showed predictive abilities, while other obesity-related indices, including BMI, BAI, and TyG index, calculated without WC, showed no predictive abilities, probably because they are not sufficient to assess abdominal fat since they do not estimate body distribution.

There are several limitations to this study. The patients were not screened according to disease duration, which may affect associations between DM and microangiopathies and macroangiopathies. However, the duration of diabetes, and especially type 2 DM, is a subjective and unreliable factor which can introduce bias in such studies. In addition, we lacked data on some important variables that can influence and may be associated with the development of PAOD, such as smoking history, exercise and vascular stiffness. Nonetheless, we believe our results highlight the importance of the effect of obesity-related indices on PAOD in patients with DM. Moreover, all of the participants in this study were Taiwanese, so our conclusions may not be generalizable to different populations. Furthermore, as this was a cross-sectional study, causal relationships and long-term clinical outcomes could not be confirmed. Further prospective studies are warranted to investigate the development and progression of PAOD among diabetic patients.

## 5. Conclusions

In conclusion, in this study, we identified associations between a low ABI and obesity-related indices, including WHtR, BRI, CI, VAI and ABSI, among patients with diabetes. Moreover, WHR, CI and LAP were the three most powerful predictors of PAOD. Clinically, WHR, CI and LAP could be used to facilitate the screening of PAOD.

## Figures and Tables

**Table 1 jpm-11-00533-t001:** Comparison of baseline characteristics between patients with ABI ≥ 0.9 and < 0.9.

Characteristics	All Patients (*n* = 1872)	ABI ≥ 0.9 (*n* = 1795)	ABI < 0.9 (*n* = 77)	*p*
Age (year)	64.0 ± 11.3	63.7 ± 11.2	71.5 ± 10.2	<0.001
Male gender (%)	43.2	43.1	44.2	0.857
Systolic blood pressure (mmHg)	134.9 ± 18.8	134.7 ± 18.8	138.9 ± 19.2	0.063
Diastolic blood pressure (mmHg)	77.9 ± 11.3	78.0 ± 11.2	75.8 ± 12.1	0.127
Body height (cm)	159.0 ± 8.4	159.0 ± 8.3	158.2 ± 9.1	0.439
Body weight (Kg)	65.5 ± 10.9	65.5 ± 10.9	66.6 ± 12.5	0.451
Waist circumference (cm)	89.6 ± 9.6	89.5 ± 9.5	93.4 ± 9.6	0.001
Hip circumference (cm)	98.5 ± 7.6	98.5 ± 7.6	98.6 ± 9.4	0.862
ABI	1.10 ± 0.11	1.12 ± 0.08	0.77 ± 0.09	<0.001
Laboratory parameters				
Fasting glucose (mg/dL)	148.6 ± 52.0	148.9 ± 52.0	142.2 ± 50.9	0.263
HbA_1c_ (%)	7.7 ± 1.7	7.7 ± 1.7	7.4 ± 1.6	0.115
Triglyceride (mg/dL)	126 (90–177)	125 (90–175)	153 (113.5–217.5)	0.002
Total cholesterol (mg/dL)	185.6 ± 38.3	185.5 ± 38.1	188.5 ± 43.2	0.562
HDL cholesterol (mg/dL)	49.7 ± 13.1	49.9 ± 13.1	45.0 ± 12.9	0.002
LDL cholesterol (mg/dL)	104.4 ± 28.2	104.4 ± 28.1	105.2 ± 32.1	0.814
eGFR (mL/min/1.73 m^2^)	68.3 ± 20.4	69.0 ± 20.2	50.6 ±19.2	<0.001
Medications				
ACEI and/or ARB (%)	73.5	72.5	97.4	<0.001
Statin use (%)	59.6	59.0	72.7	0.016
Obesity-related indices				
BMI (kg/m^2^)	25.8 ± 3.6	25.9 ± 3.6	26.5 ± 3.9	0.111
WHR	0.91 ± 0.07	0.91 ± 0.07	0.95 ± 0.07	<0.001
WHtR	0.56 ± 0.06	0.56 ± 0.06	0.59 ± 0.06	<0.001
LAP	50.5 ± 43.3	49.8 ± 42.6	66.5 ± 54.1	0.009
BRI	4.7 ± 1.7	4.7 ± 1.4	5.3 ± 1.3	<0.001
CI	1.28 ± 0.09	1.28 ± 0.09	1.33 ± 0.08	<0.001
VAI	2.6 ± 2.8	2.5 ± 2.8	3.5 ± 3.4	0.002
BAI	31.4 ± 5.2	31.3 ± 5.2	31.8 ± 5.7	0.492
AVI	16.3 ± 3.4	16.3 ± 3.4	17.7 ± 3.6	0.001
ABSI	0.082 ± 0.005	0.081 ± 0.005	0.084 ± 0.005	<0.001
TyG index	9.1 ± 0.7	9.1 ± 0.7	9.3 ± 0.7	0.090

Abbreviations. ABI, ankle–brachial index; HDL, high-density lipoprotein; LDL, low-density lipoprotein; eGFR, estimated glomerular filtration rate; ACEI, angiotensin-converting enzyme inhibitor; ARB, angiotensin II receptor blocker; BMI, body mass index; WHR, waist–hip ratio; WHtR, waist-to-height ratio; LAP, lipid accumulation product; BRI, body roundness index; CI, conicity index; VAI, visceral adiposity index; BAI, body adiposity index; AVI, abdominal volume index; ABSI, a body shape index; TyG index, triglyceride glucose index.

**Table 2 jpm-11-00533-t002:** Association of obesity-related indices with ABI using univariable linear regression analysis.

Obesity Related Indices	Univariable	
	Unstandardized Coefficient β (95% Confidence Interval)	*p*
Age (per 1 year)	−0.001 (−0.001, 0)	0.001
Male gender (vs. female)	0.006 (-0.003, 0.016)	0.194
Systolic blood pressure (per 1 mmHg)	−5.157 × 10^−5^ (0,0)	0.693
Diastolic blood pressure (per 1 mmHg)	0 (0, 0.001)	0.492
Fasting glucose (mg/dL)	3.873 × 10^−6^ (0, 0)	0.935
HbA_1c_ (per 1%)	–0.001 (−0.003, 0.002)	0.730
Triglyceride (log per 1 mg/dL)	–0.034 (−0.055, −0.014)	0.001
Total cholesterol (per 10 mg/dL)	−0.002 (−0.003, 0)	0.017
HDL cholesterol (per 1 mg/dL)	0.001 (0, 0.001)	0.001
LDL cholesterol (per 1 mg/dL)	−0.002 (−0.004, 0)	0.013
eGFR (per 1 mL/min/1.73 m^2^)	0.001 (0, 0.001)	<0.001
ACEI and/or ARB use	−0.024 (−0.035, −0.013)	<0.001
Statin use	−0.021 (−0.031, −0.012)	<0.001
BMI (per 1 kg/m^2^) *	−0.001 (−0.003, 0)	0.036
WHR (per 0.1) *	−0.011 (−0.018, −0.004)	0.003
WHtR (per 0.1) *	−0.016 (−0.024, −0.009)	<0.001
LAP (per 10) ^#^	−0.002 (−0.003, −0.001)	<0.001
BRI (per 1) *	−0.008 (−0.011, −0.004)	<0.001
CI (per 0.1) *	−0.011 (−0.016, −0.005)	<0.001
VAI (per 1) ^†^	−0.003 (−0.005, −0.001)	0.001
BAI (per 1) *	−0.001 (−0.002, −0.001)	0.002
AVI (per 1) *	−0.002 (−0.004, −0.001)	0.003
ABSI (per 0.01) *	−0.015 (−0.024, −0.006)	0.001
TyG index (per 1) ^#^	−0.009 (−0.016, −0.002)	0.016

Values expressed as unstandardized coefficient β and 95% confidence interval. Abbreviations are the same as in Table 1. * Adjusted for age, sex, log triglyceride, total cholesterol, HDL cholesterol, LDL cholesterol, eGFR and medication use. ^#^ Adjusted for age, sex, total cholesterol, HDL cholesterol, LDL cholesterol, eGFR and medication use. ^†^ Adjusted for age, sex, total cholesterol, LDL cholesterol, eGFR and medication use.

**Table 3 jpm-11-00533-t003:** Association of obesity-related indices with ABI using multivariable linear regression analysis.

Obesity-Related Indices	Multivariable	
	Unstandardized Coefficient β (95% Confidence Interval)	*p*
BMI (per 1 kg/m^2^) *	0 (−0.002, 0.001)	0.495
WHR (per 0.1) *	−0.006 (−0.014, 0.003)	0.177
WHtR (per 0.1) *	−0.009 (−0.017, 0)	0.045
LAP (per 10) ^#^	−1.636 × 10^−5^ (−0.002, 0.002)	0.986
BRI (per 1) *	−0.005 (−0.008, −0.001)	0.021
CI (per 0.1) *	−0.007 (−0.013, 0)	0.042
VAI (per 1) ^†^	−0.003 (−0.005, −0.001)	0.003
BAI (per 1) *	−0.001 (−0.002, 0)	0.052
AVI (per 1) *	−0.001 (−0.003, 0)	0.171
ABSI (per 0.01) *	−0.010 (−0.020, 0)	0.043
TyG index (per 1) ^#^	0.008 (−0.003, 0.018)	0.138

Values expressed as unstandardized coefficient β and 95% confidence interval. Abbreviations are the same as in Table 1. * Adjusted for age, sex, log triglyceride, total cholesterol, HDL cholesterol, LDL cholesterol, eGFR and medication use. ^#^ Adjusted for age, sex, total cholesterol, HDL cholesterol, LDL cholesterol, eGFR and medication use. ^†^ Adjusted for age, sex, total cholesterol, LDL cholesterol, eGFR and medication use.

**Table 4 jpm-11-00533-t004:** Area under curve (AUC), cutoff value, sensitivity, specificity and Youden index of 11 obesity-related indices for ABI < 0.9.

Obesity-Related Indices	AUC (95% Confidence Interval)	*p*	Cutoff Value	Sensitivity (%)	Specificity (%)	Youden Index
BMI (kg/m^2^)	0.547 (0.472–0.621)	0.166	25.902	54.5	53.7	0.082
WHR	0.661 (0.600–0.722)	<0.001	0.932	63.6	64.2	0.278
WHtR	0.630 (0.567–0.692)	<0.001	0.576	59.7	60.2	0.199
LAP	0.642 (0.584–0.699)	<0.001	46.050	64.9	58.4	0.233
BRI	0.630 (0.567–0.692)	<0.001	4.886	59.7	60.2	0.199
CI	0.660 (0.598–0.721)	<0.001	1.306	64.9	63.4	0.283
VAI	0.633 (0.570–0.696)	<0.001	2.219	61.0	59.3	0.203
BAI	0.527 (0.458–0.596)	0.420	31.332	51.9	55.4	0.073
AVI	0.614 (0.548–0.679)	0.001	16.625	57.1	58.4	0.155
ABSI	0.638 (0.575–0.702)	<0.001	0.083	61.0	62.9	0.239
TyG index	0.559 (0.491–0.627)	0.078	9.100	51.9	51.8	0.037

Abbreviations are the same as in Table 1.

## Data Availability

Due to restrictions placed on the data by the Personal Information Protection Act of Taiwan, the minimal data set cannot be made publicly available. Data may be available upon request to interested researchers. Please send data requests to: Szu-Chia Chen, PhD, MD. Division of Nephrology, Department of Internal Medicine, Kaohsiung Medical University Hospital, Kaohsiung Medical University.

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
