# Peer review of "Obesity-Related Indices Are Associated with Peripheral Artery Occlusive Disease in Patients with Type 2 Diabetes Mellitus"

_jpm, 2021, doi:10.3390/jpm11060533_

Round 1
Reviewer 1 Report
The Manuscript 'Obesity-Related Indices are Associated with Peripheral Artery 2 Occlusive Disease in Patients with Type 2 Diabetes Mellitus' by Wung et al. show that waist-hip ratio, Conicity Index, and Lipid accumulation product are powerful predictors of Peripheral Artery Occlusive Disease.
There are 1064 female patients enrolled in the study. However, the authors present data only on the male population. Is there a specific reason? Do the parameters predict PAOD only in males or females also?
Do the predictors correlate with the severity of PAOD?
Author Response
The Manuscript 'Obesity-Related Indices are Associated with Peripheral Artery 2 Occlusive Disease in Patients with Type 2 Diabetes Mellitus' by Wung et al. show that waist-hip ratio, Conicity Index, and Lipid accumulation product are powerful predictors of Peripheral Artery Occlusive Disease.
- There are 1064 female patients enrolled in the study. However, the authors present data only on the male population. Is there a specific reason? Do the parameters predict PAOD only in males or females also?
Ans: Sorry for making your misunderstanding. In Table 1, we showed the percentage of male was 43.2%. However, further statistical analysis (Table 2-4) included male and female patients, not just male patients.
- Do the predictors correlate with the severity of PAOD?
Ans: Thank you for your question. An ABI , 0.9 has been used to identify PAOD in clinical practice and epidemiologic studies. Moreover, lower ABI is associated with generalized atherosclerosis. Previous studies had showed that there was a significant correlation between continuous ABI and common carotid artery intima media thickness and the degree of stenosis in the intracranial internal carotid artery and middle cerebral artery. Measurement of continuous ABI and carotid intima-media thickness might provide good prognostic atherosclerosis markers for stroke. Therefore, survey cut-off ABI < 0.9 and continuous ABI is of equal importance. In our study, except ABI < 0.9 (table 1 and 4), we also surveyed the determinants of continuous ABI (Table 2 and 3), which correlate the severity of PAOD.

Reviewer 2 Report
Dear Authors,
Thank you for sending your manuscript for review. The manuscript entitled Obesity-Related Indices are Associated with Peripheral Artery 2 Occlusive Disease in Patients with Type 2 Diabetes Mellitus is very interesting and concerns a clinically significant problem.
Manuscrpit written well, with all the rules for writing work. However, I have a few weak points:
1. Introduction - written in an interesting way, I think it is worth it in the paragraph in which the anthropometric indices are listed, more citations, not only Cheng C.
2. The aim is not clearly specified. Please provide the general goal and if these are the specific goals of the research.
3. Material and methodology
- the criteria for exclusion from the study should be detailed - whether the study included patients with neoplastic disease, with previously diagnosed chronic ischemia of the lower limbs, patients with heart failure.
- an important element is the age of the patients included in the study - whether age was the inclusion criterion.
- the authors focused on renal failure, however, diabetes causes multi-organ complications - why others were not discussed
- please describe whether the test was performed once and by whom, under what conditions the test was performed.
The results
- it is surprising that such a small percentage of patients obtained the result above 0.9. Is the ABI result correlated with the age of patients?
Why critical ischemia and abnormal vascular stiffness were not taken into account and is common in diabetes mellitus.
Figure 3 is hardly readable, in table 4, I propose to add statistical significance as another column because the results are not readable.
Conclusions
Unfortunately they do not bring revealing elements in knowledge. We know that obesity and diabetes significantly determine the ischemia of the lower extremities, so all elements related to carbohydrate metabolism disorders and the accumulation of adipose tissue affect PAOD.
Author Response
Dear Authors,
Thank you for sending your manuscript for review. The manuscript entitled Obesity-Related Indices are Associated with Peripheral Artery 2 Occlusive Disease in Patients with Type 2 Diabetes Mellitus is very interesting and concerns a clinically significant problem. Manuscript written well, with all the rules for writing work. However, I have a few weak points:
- Introduction - written in an interesting way, I think it is worth it in the paragraph in which the anthropometric indices are listed, more citations, not only Cheng C.
Ans: Thank you for your suggestion. We have added more citations from reference 13 to 15. (Line 65)
- The aim is not clearly specified. Please provide the general goal and if these are the specific goals of the research.
Ans: Thank you for your comment. We only explained how we conducted the study except the aim of our study in final paragraph of introduction. We have added “The aim of our study is to reveal the association between obesity and PAOD through obesity-related indices among diabetic population.”
- The aim of this study was to examine the association between obesity and PAOD through obesity-related indices among a diabetic population. In this study, we enrolled 1872 diabetic patients residing in the south of Taiwan and investigated the associations between PAOD defined according to ABI and the aforementioned obesity-related indices. (Line 73-77)
- Material and methodology
- the criteria for exclusion from the study should be detailed - whether the study included patients with neoplastic disease, with previously diagnosed chronic ischemia of the lower limbs, patients with heart failure.
Ans: Thank you for your comments. In our study, except excluding 1) dialysis; 2) renal transplantation; and 3) type 1 DM, we also excluded patients with neoplastic disease and critical ischemia condition. We have added in the methods.
- The following patients were excluded: 1) those who were receiving dialysis; 2) whose who had had undergone a renal transplantation; 3) those with type 1 DM (defined as the continued need for insulin treatment for at least 1 year after the diagnosis, the presence of diabetic ketoacidosis, ketonuria [³ 3], or symptoms of severe hyperglycemia), 4) those who had neoplastic disease; and 5) those with critical ischemia conditions, such as pain, paralysis, paresthesia, pulseless, and pale. (Line 81-86)
- an important element is the age of the patients included in the study - whether age was the inclusion criterion.
Ans: Thank you for your comments. We enrolled patients aged more then 18 years old. We have added in the methods.
- All outpatients aged more than 18 years with type 2 DM who visited two hospitals in southern Taiwan were enrolled in this study. (Line 80-81)
- the authors focused on renal failure, however, diabetes causes multi-organ complications - why others were not discussed
Ans: Thank you for your comments. According to our previous studies in CKD patients (Kaohsiung J Med Sci. 2009 Jul;25(7):366-73), because renal failure may influence the ABI data, therefore, in our study, we excluded dialysis and renal transplantation. Besides, we also excluded patients with neoplastic disease and critical ischemia condition.
- please describe whether the test was performed once and by whom, under what conditions the test was performed.
Ans: Thank you for your suggestion. We have added the condition the test was performed.
- ABI was measured once in each patient using a non-invasive vascular screening device (VP1000; Collin Co. Ltd., Komaki, Japan) at the diabetes out-patient clinics by a trained diabetic nurse [21-23]. Before ABI was measured, study patients were instructed to lie quietly and breathe normally in the supine position for at least 10 minutes. (Line 107-110)
- The results
- it is surprising that such a small percentage of patients obtained the result above 0.9. Is the ABI result correlated with the age of patients?
Ans: Thank you for your comments. In our study, 4.1% of the enrolled patients had an ABI < 0.9, which was lower compared with previous studies. Many factors may influence the difference, such as age, ethnicity, cormorbidity, medications and life style. In our study, we can see that old age is associated with low ABI in Table 2. Further studies maybe needed to investigate the possible causes.
- Why critical ischemia and abnormal vascular stiffness were not taken into account and is common in diabetes mellitus.
Ans: Thank you for your comments. We totally agreed that critical ischemia and abnormal vascular stiffness may influence PAOD in DM population. We have added critical ischemia in exclusion criteria. However, we lacked the data of pulse wave velocity in the study. We have added in the Limitation.
- In addition, we lacked data on some important variables that can influence and may be associated with the development of PAOD, such as smoking history , exercise and vascular stiffness. (Line 260-262)
- Figure 1 is hardly readable, in table 4, I propose to add statistical significance as another column because the results are not readable.
Ans: Thank you for your comments. We have deleted Figure 1, and added statistical significance in the Results and Table 4.
- The AUCs for the indices in descending order were WHR (0.661, p < 0.001), CI (0.660, p < 0.001), LAP (0.642, p < 0.001), ABSI (0.638, p < 0.001), VAI (0.633, p < 0.001), WHtR and BRI (0.630, p < 0.001), AVI (0.614, p = 0.001), TyG index (0.559, p = 078), BMI (0.547, p = 0.166) and BAI (0.527, p = 0.420). (Line 170-174)
- Conclusions
Unfortunately they do not bring revealing elements in knowledge. We know that obesity and diabetes significantly determine the ischemia of the lower extremities, so all elements related to carbohydrate metabolism disorders and the accumulation of adipose tissue affect PAOD.
Ans: Thank you for your comments. Although our results do not bring revealing elements in knowledge. Nonetheless, we believe our results highlight the importance of the effect of obesity-related indices on PAOD in patients with DM.

Round 2
Reviewer 1 Report
The authors have addressed my concerns.
Reviewer 2 Report
Dear Editor,
the authors of the paper addressed all comments, clarifying all doubts.